# Positivity Trends of Bacterial Cultures from Cases of Acute and Chronic Periprosthetic Joint Infections

**DOI:** 10.3390/jcm11082238

**Published:** 2022-04-16

**Authors:** Rares Mircea Birlutiu, Cristian Ioan Stoica, Octav Russu, Razvan Silviu Cismasiu, Victoria Birlutiu

**Affiliations:** 1Foisor Clinical Hospital of Orthopedics, Traumatology, and Osteoarticular TB, B-dul Ferdinand 35–37, Sector 2, 021382 Bucharest, Romania; 2Foisor Clinical Hospital of Orthopedics, Traumatology, and Osteoarticular TB, Carol Davila University of Medicine and Pharmacy, B-dul Ferdinand 35–37, Sector 2, 021382 Bucharest, Romania; cristi.stoica@foisor.ro (C.I.S.); razvan.cismasiu@gmail.com (R.S.C.); 3Department of Orthopaedics and Traumatology, Clinical County Hospital, George Emil Palade University of Medicine, Pharmacy, Science, and Technology of Targu Mures, Str. Gheorghe Marinescu 38, 311694 Targu Mures, Romania; octav@genunchi.ro; 4Faculty of Medicine, Lucian Blaga University of Sibiu, Str. Lucian Blaga, Nr. 2A, 550169 Sibiu, Romania; victoriabirlutiu@yahoo.com

**Keywords:** periprosthetic joint infection, PJI, diagnosis, acute, chronic, positivity trends, bacterial cultures

## Abstract

Background: There is no clear distinction in the literature regarding the positivity trends of bacterial cultures in acute and chronic prosthetic joint infections. Methods: We prospectively included in this study all consecutive patients, aged over 18 years, that were hospitalized from September 2016 through December 2019, that underwent a joint arthroplasty revision surgery. Results: Forty patients were included in our analysis, 11 acute/acute hematogenous and 29 chronic PJIs. We were able to identify all strains of acute/acute hematogenous PJIs within 3 days, whereas this took 8 days for chronic PJIs. Sonication fluid cultures increased the positivity rate and helped in identifying rare pathogens such as *Ralstonia pickettii* from chronic PJIs, but also increased the number of identified strains from acute PJIs. Culturing synovial fluid in our study did not seem to have a clear benefit compared to sonication fluid and periprosthetic tissue cultures. Conclusion: There was a different positivity trend in bacterial cultures. Empiric broad-spectrum antibiotic therapy can be re-evaluated after 3 days for acute PJIs. A prolonged incubation time, especially in the case of chronic PJIs, is mandatory; however, extending the incubation period beyond 14 days would not further improve the ability to identify microorganisms.

## 1. Introduction

A broad range of infections are biofilm-related infections (BRI), from catheter-associated urinary tract infections (which still represent the most common BRIs) to central line-associated bloodstream infections, fracture-related infections, BRI associated with the use of fixed braces [1,2], and periprosthetic joint infections [3]. Periprosthetic joint infections (PJIs) are devastating complications following total joint arthroplasty, most commonly associated with total hip or knee arthroplasty (due to the increased number of this type of surgery), with broad implications, and with significant morbidity and mortality [4]. A total of 16.8% of all knee-revision surgeries and 14.8% of all hip-revision surgeries are due to failure caused by PJIs [5]. Prosthetic joint infections occur at a frequency of 1% to 3% and are still a major cause of healthcare expenditure [6].

Detection of etiological pathogens represents a key aspect of adequate therapy for periprosthetic joint infections. Unfortunately, it is reported that in up to 50% of periprosthetic joint-infection cases, the identification of the causative organism fails [7]. There has been constant debate regarding the length of incubation of prosthetic joint-infection samples. In 2013, at the International Consensus Meeting, meeting delegates agreed that cultures should be incubated for 5–14 days. The 2018 International Consensus Meeting for Periprosthetic Joint Infection recommended that cultures should be maintained for a period of 5 to 7 days, and when periprosthetic joint infections with low-virulence organisms are suspected, or if preoperative cultures are negative but there is a high clinical suspicion, that the cultures should be maintained for up to 21 days [8,9]. Virulent bacteria have a faster replication rate, whereas less virulent bacteria have a slower replication rate [10]. In the case of biofilm-related infections, the biofilm type plays an important role in the replication rate, and mature biofilms are more difficult to cultivate [11].

As there is no clear distinction and there are few data in the literature regarding the positivity trends of bacterial cultures between acute and chronic prosthetic joint infections, in this study we report our results from a single-center observational study to describe the positivity trend of bacterial cultures from different specimens from cases of acute and chronic periprosthetic joint infections. As a secondary aim, we also assessed from the VITEK 2 Compact analyzer (bioMérieux, Marcy-l’Étoile, France) and reported the analysis time of the tested isolates.

## 2. Materials and Methods

### 2.1. Study Design

We conducted a single-center observational, cohort, ongoing study in a county hospital with 1054 beds: the Academic Emergency Hospital Sibiu, Romania. Before we included the patients in the study, the study protocol was reviewed and approved by the institutional review board. A standardized diagnostic system was used to assess all patients who underwent a surgical intervention for the revision of a joint prosthesis to determine implant failure. Our implemented diagnostic strategy included a sampling of at least four intraoperative tissue specimens (one sample of the periprosthetic membrane was used for the histopathological examination), and at least three samples were sent to the laboratory for bacterial cultures. Sonication of the removed orthopedic prosthetic components and harvesting of the sonication fluid for culture, incubation, and cell counting of the synovial fluid were performed. As a rapid method of bacteria detection from the sonication fluid, we used a bbFISH kit (hemoFISH Masterpanel, Miacom diagnostics GmbH Düsseldorf, Germany). All specimens were inoculated on aerobic and anaerobic culture media, and a 14-day period of incubation period was implemented.

### 2.2. Study Population

We prospectively included all consecutive patients, aged over 18 years, who were hospitalized from September 2016 through December 2019 and underwent joint arthroplasty-revision surgery for any reason. All cases of culture-negative periprosthetic joint infections were excluded from this analysis. In addition, we excluded bacterial cultures from reimplantation surgeries and the sonication fluid cultures from spacer sonication. Detailed information was extracted from the medical records of the patients using a standardized collection form. All data were available for all the enrolled patients.

### 2.3. Laboratory Studies

Our newly implemented diagnostic included a standardized sampling of at least four intraoperative tissue samples (one of the samples used for the histopathological examination (periprosthetic membrane) and the other were sent to the microbiological laboratory for bacterial cultures). For the sonication of the retrieved implants, in the operating theater, sterile Ringer’s or saline solution was added over the implants that were deposited in sterile containers. Containers were previously sterilized according to the manufacturer’s instructions and double-packed. The implants were processed within 30 min by sonication (1 min) using an ultrasound bath (BactoSonic14.2, Bandelin GmbH, Berlin, Germany) at a frequency of 42 kHz and a power density of 0.22 W/cm^2^. The resulting sonication fluid was vortexed, and 50 mL of sonication fluid was centrifuged at 2500 rpm for 5 min. The resulting precipitate was inoculated. If >50 CFU/mL was counted, sonication fluid cultures were considered positive. Ten milliliters of sonication fluid were incubated in blood-culture bottles in a blood-culture system (BD BACTEC™, New Jersey, USA). Regarding the periprosthetic tissue cultures, tissue samples were collected in sterile vials and individually homogenized in 1 mL thioglycolate broth. Tissue homogenate samples (1 mL) were inoculated into the culture media. Synovial fluid was aspirated preoperatively in a native vial and inoculated into different media for culturing. All biological samples that required cultures were inoculated and incubated aerobically, anaerobically, and in a high concentration of CO_2_ (GENbag-GENbox Atmospheric generators bioMérieux, Marcy-l’Étoile, France) at 37 °C. The isolated bacteria were identified using a VITEK 2 Compact analyzer (bioMérieux, Marcy-l’Étoile, France). Minimum inhibitory concentrations were assessed according to the European Committee on Antimicrobial Susceptibility Testing breakpoints. We also assessed the analysis time from the VITEK 2 Compact analyzer (bioMérieux, Marcy-l’Étoile, France) report. A 14-day period of incubation was implemented. We were able to analyze cultures during working days and weekends. We previously published full details of the implemented protocol in some articles [12,13].

### 2.4. Study Definitions and Classification

A culture was marked as positive on the day that an isolate was identified by the VITEK 2 Compact analyzer (bioMérieux, Marcy-l’Étoile, France), the first day of growth. We considered a culture to be truly positive (and as a confirmed PJI) according to the MSIS criteria (i.e., two positive cultures with the same microorganism or one positive culture and a combined preoperative and postoperative score ≥ 4 (elevated serum CRP (mg/L) or D-Dimer (μg/L); elevated serum ESR (mm/h); elevated synovial WBC (cells/μL) or leukocyte esterase; elevated synovial PMN (%); positive histology; positive intraoperative purulence) as the presence of a single positive culture represents a score of 2 points (the total combined preoperative and postoperative score should ≥ 6 for a positive diagnosis of PJI). [14,15].

Periprosthetic joint infection was defined using the criteria from the Musculoskeletal Infection Society workgroup published by Parvizi et al. [14]. We used the classification proposed by Zimmerli et al. to determine if there was an acute, late chronic, or acute late periprosthetic joint infection, a classification that defines the prosthetic joint infections as early (occurring within 3 months after surgery), delayed (3–24 months) and late (>24 months) [6]. Due to the low number of enrolled patients, we also used a much simpler classification, a classification from the Pocket Guide to Diagnosis & Treatment of Periprosthetic Joint Infection (PJI) of the PRO-IMPLANT Foundation, Berlin, Germany (coordinated by N. Renz and A. Trampuz), a guide that is in line with national and international recommendations and that defines periprosthetic infections as acute or chronic (Perioperative/hematogenous or per continuitatem).

### 2.5. Statistical Analysis

We performed the statistical analysis using the IBM SPSS Statistics^®^ (Chicago, IL, USA) version 28 software. Categorical variables, expressed as counts and percentages, were analyzed using the Chi-square and Fisher’s exact tests. Continuous variables were described as means (standard deviation) and medians. The assessment of differences in distributions between groups was performed using the Mann–Whitney nonparametrical test. A *p*-value of 0.05 or less was considered significant.

## 3. Results

From September 2016 to December 2019, 84 patients underwent a debridement, antibiotics, and implant retention procedures or one-stage or two-stage revision surgery. A diagnosis of aseptic loosening of an endoprosthetic implant was established in 30 patients. A total of 51 cases were culture positive. We excluded from the final analysis 5 cultures that were considered contaminants and 6 cases with positive cultures from the second stage of 2-stage revision surgery. Three patients had culture-negative PJIs. A total of 40 confirmed PJIs were included in our final analysis. Of the 40 cases analyzed, 11 had acute/acute hematogenous PJIs and 29 had chronic PJIs. A total of 7 cases were acute PJIs and 4 were acute hematogenous PJIs. Due to the low number of enrolled patients, we decided to analyze the acute and acute hematogenous PJIs together. The 40 cases included 17 hip prosthesis and 23 knee prosthesis. We were able to isolate 11 microorganisms from the 11 acute PJI cultures. The 29 chronic PJI cultures yielded 33 microorganisms.

We were able to group the 40 patients diagnosed with a periprosthetic joint infection, using the classification proposed by Zimmerli et al., as follows: 10 patients diagnosed with early PJI, 7 patients with delayed PJI, and 23 patients were diagnosed with a late PJI.

Using the classification of the periprosthetic joint infections proposed in the Pocket Guide by the PRO-IMPLANT Foundation, 7 patients were diagnosed with an acute perioperative infection, 4 patients with acute hematogenous infection, and 29 patients with chronic PJI. We report all of our results using this classification. (Figure 1).

From all periprosthetic joint-infection cultures that we analyzed, 44 unique strains that were considered noncontaminants were identified. A culture was marked as positive on the day that an isolate was identified by VITEK 2 Compact analyzer (bioMérieux, Marcy-l’Étoile, France).

Detailed information regarding the types of isolated strains from different sample types (sonication fluid cultures, periprosthetic tissue cultures, and synovial fluid cultures) are reported in Table 1, Table 2 and Table 3.

For the isolated strains from acute PJIs bacterial cultures (n = 11), the median time until the cultures became positive is 1.5 days (range 1–3, mean 1.5 ± 0.68). We were able to identify 90% of the bacterial species within 2 days after inoculation of the specimens from different clinical samples and 100% within 3 days. For acute PJIs, sonication fluid and periprosthetic tissue cultures showed a similar positivity trend: 100% of the cultures from the sonication fluid were positive by the 2nd day and 83% of tissue cultures by the 2nd day. However, we were able to identify by culturing the sonication fluid with 5 additional microorganisms, as compared to periprosthetic tissue cultures. Culturing synovial fluid in our study did not seem to have a clear added benefit (only 3 strains were identified) compared to the sonication fluid and periprosthetic tissue cultures. (Figure 2.)

For the strains isolated from chronic PJIs bacterial cultures (n = 33), the median time until the cultures became positive was 3 days (range 1–8, mean 2.83 ± 1.57). We were able to identify 78% of the bacterial species within 3 days after inoculation of the specimens from different clinical samples and 100% within 8 days. For chronic PJIs, sonication fluid and periprosthetic tissue cultures showed a similar positivity trend: 90% of the cultures from the sonication fluid were positive by the 4th day and 100% of the tissue cultures were also positive by the 4th day. However, we were able to identify by culturing the sonication fluid with additional microorganisms as compared to periprosthetic tissue cultures. 18 additional strains were identified by sonication fluid cultures when compared with periprosthetic tissue cultures. Culturing synovial fluid from chronic PJIs in our study did not seem to have a clear added benefit (only 5 strains were identified) compared to the sonication fluid and periprosthetic tissue cultures. In chronic PJIs, periprosthetic tissue cultures had the fastest culture yield (87% after 3 days), but only from the sonication fluid cultures were we able to identify the following strains: *Enterobacter amnigenus* 2, *Klebsiella* spp., *Pseudomonas fluorescens*, and *Ralstonia pickettii* (4 strains). We were able to identify two polymicrobial chronic PJIs (*Ralstonia pickettii* + *Pseudomonas aeruginosa*, and *Staphylococcus xylosus* + *Acinetobacter* spp.) only from sonication fluid cultures. (Figure 3.)

The time to positivity was lower in acute periprosthetic joint-infection bacterial cultures compared to chronic periprosthetic joint-infection cultures. Sonication fluid cultures increased the positivity rate and might be the only way of identifying some pathogens.

As the number of positive bacterial cultures from the sonication fluid was larger than that from other types of cultures, we decided to additionally analyze the time to positivity in relation to the type of pathogen isolated by sonication fluid cultures. There is a clear difference in the positivity trends between more virulent species such as *Staphylococcus aureus*, and less virulent species such as CoNS—coagulase-negative staphylococci, *Ralstonia pickettii*, and aerobic Gram-negative rods (Figure 4). *Ralstonia pickettii* showed the longest time to positivity, growing from the 4th day on and taking 8 days to isolate all strains. More than 80% of virulent organisms were isolated on days 1 or 2 after incubation. The coagulase-negative staphylococci group showed approximately 77% positivity in the first 2 days. It took 4 days to isolate all coagulase-negative staphylococci (2 days in acute PJI, 4 days in chronic PJI). Aerobic Gram-negative rods showed a linear positivity trend curve with a maximum of 8 days to isolate all species.

The positivity trend of the cultures was dependent on the prosthetic joint-infection-causing species.

It has already been mentioned in the literature that biofilm formation is closely correlated with the time a periprosthetic joint infection exists; in this context, we decided to analyze the correlation between the time to positivity from the sonication fluid cultures and the age of the implant. For all samples tested from acute PJIs, we did not observe a correlation between the two parameters (*p*-0.66, Mann–Whitney U test). When we analyzed the same data from chronic PJIs, we observed a correlation between the time to positivity from the sonication fluid cultures and the age of the implant (*p*-0.03, Mann–Whitney U test).

One of the parameters that can be tracked from the laboratory report and that is evaluated by the VITEK 2 Compact (bioMérieux, Marcy-l’Étoile, France), is the analysis time, which represents the period in which the bacterial identification and antibiotic susceptibility tests were performed. We analyzed this parameter from sonication fluid cultures for the most commonly identified strains. The analysis time was a specific parameter for each bacterial strain. Thus, for *Staphylococcus epidermidis* strains, the mean analysis time was 10.75 h with a standard deviation of ±1.20 h; in the case of Methicillin-resistant *Staphylococcus aureus* strains, the mean analysis time was 11.88 h with a standard deviation of ±0.67 h; and in the case of *Ralstonia pickettii* strains, the average duration of analysis was 17.50 h with a standard deviation of ±1.39 h; also in the case of *Ralstonia pickettii* strains, the longest analysis time was identified as 19.75 h.

## 4. Discussion

Periprosthetic joint infections remain an issue and are the most feared complication associated with joint arthroplasty surgeries. The diagnosis and management of PJIs require adapted protocols for the treatment of biofilm-related infections and new and rapid diagnostic methods to be able to improve the eradication rate. However, there is no 100% certainty that the infection was eradicated [16].

A prolonged incubation time for bacterial cultures from PJIs of 10 to 14–21 days is generally recommended in the literature and is needed; unfortunately, until this moment there are few studies that attempted to make a clear distinction between acute and chronic PJIs. There are data in the literature that recommend a prolonged period of incubation for up to 21 days to minimize the culture-negative PJI, especially if until the 14 days no microorganism has been isolated [17].

In this study, we retrospectively analyzed prospectively collected data regarding positivity trends in acute, acute hematogenous, and chronic infections, and reported that the time to positivity of acute/acute hematogenous PJI cultures is definitely shorter compared to chronic PJI cultures. In our analysis, we were able to identify all strains of acute/acute hematogenous PJIs within 3 days, whereas this took 8 days for chronic PJIs. Sonication fluid cultures increase the positivity rate and might be the only way to identify some pathogens, especially from chronic PJIs, but also to increase the number of identified strains from acute PJIs. Culturing synovial fluid in our study did not seem to have a clear added benefit compared to the sonication fluid and periprosthetic tissue cultures for both acute and chronic PJIs.

Talsma et al. reported in their study that all isolates of acute PJIs grew within 5 days, while this took 11 days for chronic PJIs, a slightly longer incubation period than ours. In contrast to our results, they reported that sonication showed no benefit concerning time to positivity for acute PJIs [18]. Klement et al. reported that in their study, extending the duration of incubation period beyond 5 days in acute hip and knee PJI did not increase the overall culture yield or surgical-treatment success [4].

The results of our study demonstrated that extending the duration of incubation did not increase the overall culture yield for acute PJIs; however, in our opinion this is mandatory for chronic PJIs. Similar recommendations were also made by Talsma et al. [18]. Schaffer et al. recommended that microbiological cultures should be incubated for 14 days after they conducted a large study with prospectively collected data. They reported a median time to diagnosis of only 4 days, with a range of up to 13 days, highlighting also the polymicrobial nature of PJIs [10]. Similar results were obtained in this study.

The presence of biofilm also in acute PJIs may influence and reduce the bacterial multiplication rate and modify the positivity trend of the cultures. It also makes sense that a shorter incubation period is needed for acute PJIs bacterial cultures to become positive due to the high bacterial load and the increased presence of planktonic bacteria [18]. Nevertheless, we should take into consideration the fact that acute PJIs may involve low-virulence microorganisms as well, and a shorter incubation period in these cases might lead to failure of the diagnosis and the management of the case [19,20]. Nevertheless, our results clearly show that there are distinct positivity patterns for acute and chronic PJIs and indicate that a prolonged incubation period might not be necessary for acute PJI specimens.

Our results and similar published data might have the potential to reduce the workload in the laboratory, and subsequently reduce diagnostic costs and the period in which broad-spectrum empiric antibiotic treatment regimens are used to decrease the risk of increasing the resistance of bacteria to antibiotics [18].

In a recent publication, Deroche et al. recommended that empiric antibiotics for PJI can also be reevaluated on day 5 in the case of polymicrobial infections [21]. Our data demonstrate that the day 5 mark indicated by Deroche et al. is optimal for acute infections, but for chronic PJIs in the context of low-virulence microorganisms, only 90% of pathogens are isolated until day 5. Similar data have been published by other authors [22].

The similarity between our data and those of Schwotzer et al. [23] taking into account that they reported only periprosthetic tissue-culture data, is the proportion of low-virulence microorganisms, which accounted for approximately 50% of infections in both study populations. They concluded that a prolonged 14-day incubation period could be of interest in the context of increased prevalence of low-virulence microorganisms and anaerobes.

In our opinion, as *R. pickettii* was isolated from the sonication fluid, there was no contamination in these cases. As a Gram-negative bacillus, nonfermentative, oxidase-positive bacteria, it can be confused with other bacteria such as *P. aeruginosa*, the isolated strain and a sample of the sonication fluid were shipped to a reference laboratory where our results were confirmed (*R. pickettii* was isolated). Nevertheless, contamination of the container in which the implant is shipped for sonication and then to the laboratory is not possible, as the implants are inserted into the container under strict sterile conditions. Ringer or saline sterile solution is added in the same manner; the next time the container will be open is at the laboratory where sonication fluid is harvested for the cultures; the container is opened in a biological safety cabinet, and the operating rooms where joint arthroplasties are performed are equipped with high-efficiency air filtration systems, based on the fact that regular epidemiological surveillance is conducted in the operating theater with samples that are collected from the surfaces, appliances, air filters, solutions, and so on. In this setting, the PJIs could be associated with the saline lavage used during the initial total joint arthroplasty—on this matter, regular samples from these solutions are harvested as means of prevention; unfortunately from the period of the TJA, there is no data on this matter. There are few reports of PJIs caused by *R. pickettii* published in the literature, but based on the above statements, as well as our previously published experience, from our results there is no doubt that *R. pickettii* is the causative agent of the PJIs.

Nevertheless, our data show that sonication of implant resulted in the isolation of unique strains in both monomicrobial and polymicrobial infections, e.g., coagulase-negative staphylococci and low-virulence Gram-negative bacilli (*Ralstonia pickettii*); failure to isolate this pathogen has major consequences in the long term. In our opinion, sonication of the implant should be performed despite the presence of acute or chronic PJIs. Here, we report a clear benefit of sonication in patients with chronic PJI samples. Being able to analyze the cultures during the weekend prevented us from having potentially biased data regarding the time to positivity in these samples. There are no published data found regarding the analysis time from the VITEK 2 Compact (bioMérieux, Marcy-l’Étoile, France) report, and based on our study and previous personal observations, the analysis time is a specific parameter of each bacterial strain, and low-virulence microorganisms tend to have an increased analysis time.

The current study also has some limitations. The main limitation of our study is the sample size of enrolled patients, which prevents us from taking firm conclusions, especially on proposing new incubation-length periods in chronic PJIs. Nevertheless, the number of enrolled patients was sufficiently high to propose as a general recommendation the use of sonication regardless of the presence of an acute or chronic prosthetic joint infection. This was a monocentric, observational, retrospective cohort study. The center where this study was conducted was not a dedicated center for the treatment of PJIs. As with any culture study, a possibility exists that the isolated strains were secondary to contamination. Larger studies are needed to confirm our results; nevertheless, they are very promising.

## 5. Conclusions

In conclusion, our results demonstrate that there is a different positivity trend for bacterial cultures in acute and chronic prosthetic joint infections. In our analysis, we were able to identify all strains of acute/acute hematogenous prosthetic joint infections within 3 days. In our study, chronic prosthetic joint-infection cultures require 8 days of incubation to reach a 100% yield. Our results indicate that empiric broad-spectrum antibiotic therapy can be re-evaluated after 3 days for acute prosthetic joint infections. Our results should be confirmed in a larger cohort of patients. A prolonged incubation time, especially in the case of a chronic prosthetic joint infection, is mandatory, but extending the incubation period beyond 14 days would not further improve the ability to identify microorganisms.

## Figures and Tables

**Figure 1 jcm-11-02238-f001:**
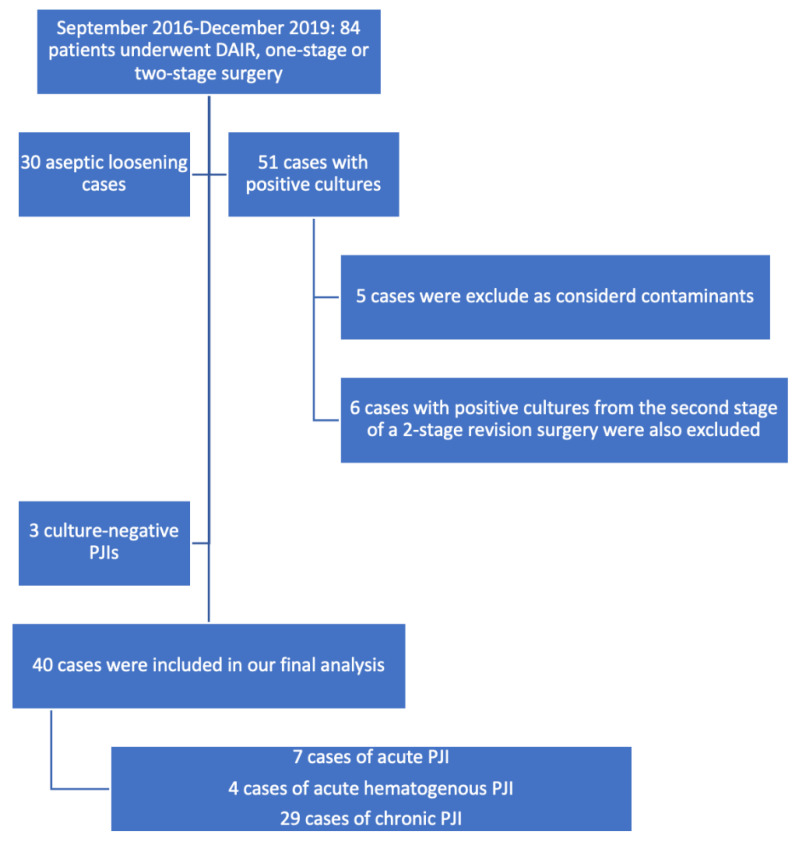
Flow diagram showing details of the enrolled patients.

**Figure 2 jcm-11-02238-f002:**
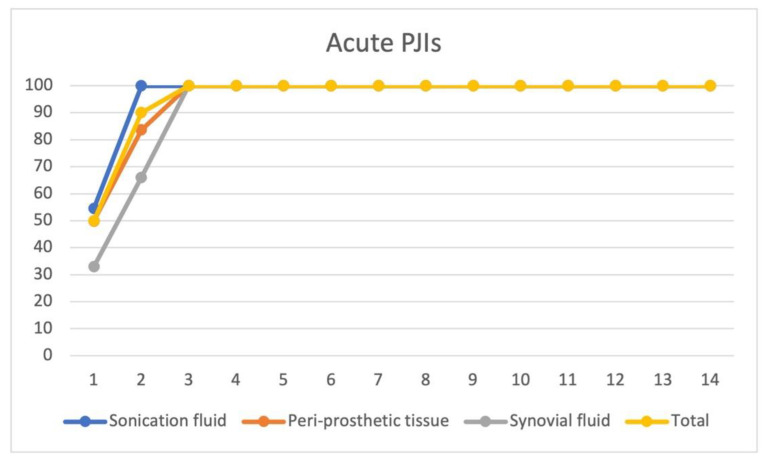
Time to positivity for bacterial cultures for acute PJI cultures. Time to positivity plotted against the percentage of culture positivity in acute (n = 11 patients) PJIs as a total (from all clinical samples), and from different clinical samples (sonication fluid cultures, periprosthetic tissue cultures, and synovial fluid cultures). On the vertical axis—culture positivity (%), and on the horizontal axis—time to positivity (days).

**Figure 3 jcm-11-02238-f003:**
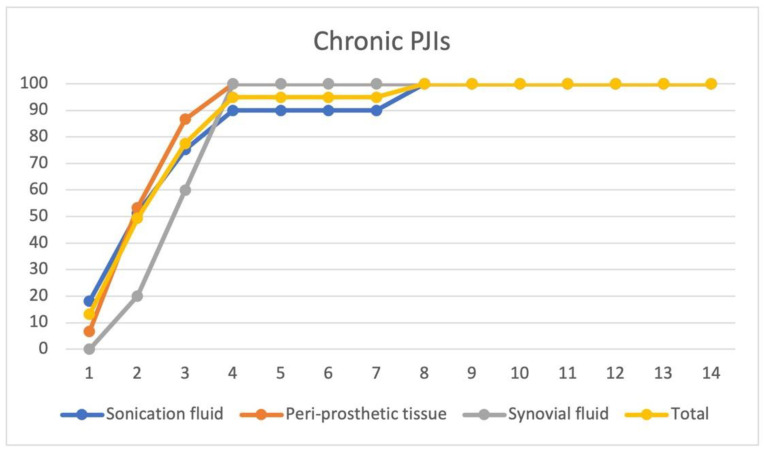
Time to positivity for bacterial cultures for chronic PJI cultures. Time to positivity plotted against the percentage of culture positivity in chronic (n = 33 patients) PJIs as a total (from all clinical samples), and from different clinical samples (sonication fluid cultures, periprosthetic tissue cultures, and synovial fluid cultures). On the vertical axis—culture positivity (%), and on the horizontal axis—time to positivity (days).

**Figure 4 jcm-11-02238-f004:**
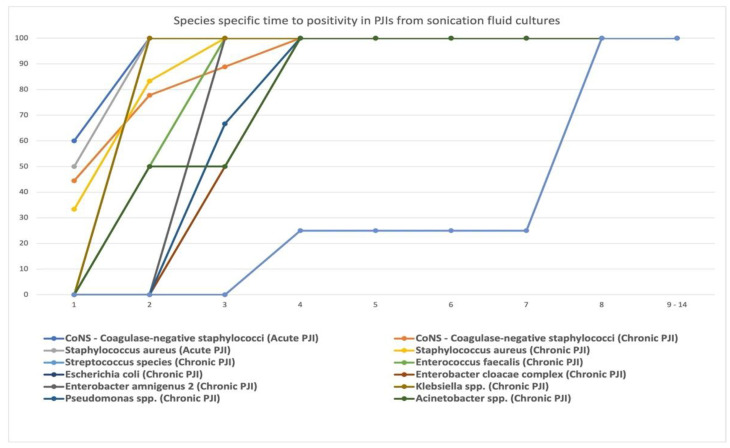
Species-specific time to positivity in PJIs from sonication fluid cultures. On the vertical axis—culture positivity (%), and on the horizontal axis—time to positivity (days).

**Table 1 jcm-11-02238-t001:** Isolated bacterial species (no. of strains identified) from sonication fluid/incubation period days.

Microorganisms	No. of Positive Cultures (No. of Strains Identified) from Sonication Fluid/Incubation Period Days						
	D 1	D 2	D 3	D 4	D 5–D 7	D 8	D 9–D14
*S. epidermidis*	3 + **3**	1 + **2**	0	0	0	0	0
*S. lentus*	1	1	0	1	0	0	0
*S. xylosus*	0	1	1	0	0	0	0
MRSA	1 + **3**	2 + **2**	0	0	0	0	0
MSSA	1	1 + **1**	1	0	0	0	0
*Str.* group D	0	1	1	0	0	0	0
*Ent. faecalis*	0	1	1	0	0	0	0
*E. coli*	0	1	0	0	0	0	0
*Enterobacter cloacae complex*	0	0	1	1	0	0	0
*Enterobacter amnigenus 2*	0	0	1	0	0	0	0
*Klebsiella* spp.	0	1	0	0	0	0	0
*P. fluorescens*	0	0	1	0	0	0	0
*P. aeruginosa*	0	0	1	1	0	0	0
*Acinetobacter* spp.	0	1	0	1	0	0	0
*Ralstonia pickettii*	0	0	0	1	0	3	0

Red—bacterial strains from acute/acute hematogenous PJIs; D—days; *S.*—*Staphylococcus*; MRSA—Methicillin-resistant *S. aureus*; MSSA—Methicillin-susceptible *S. aureus*; *Str.*—*Streptococcus*; *Ent.*—*Enterococcus*; *E.*—*Escherichia*; *P.*—*Pseudomonas*.

**Table 2 jcm-11-02238-t002:** Isolated bacterial species (no. of strains identified) from periprosthetic tissue/incubation period days.

Microorganisms	No. of Positive Cultures (No. of Strains Identified) from Periprosthetic Tissue/Incubation PERIOD Days						
	D 1	D 2	D 3	D 4	D 5–D 7	D 8	D 9–D 14
*S. epidermidis*	0	1 + **1**	** 1 **	0	0	0	0
*S. lentus*	0	0	0	1	0	0	0
*S. xylosus*	0	0	1	0	0	0	0
MRSA	1 + **2**	1 + **1**	1	1	0	0	0
MSSA	** 1 **	1	0	0	0	0	0
*Str.* group D	0	1	0	0	0	0	0
*Ent. faecalis*	0	1	1	0	0	0	0
*E. coli*	0	1	0	0	0	0	0
*Enterobacter cloacae complex*	0	0	1	0	0	0	0
*Enterobacter amnigenus 2*	0	0	0	0	0	0	0
*Klebsiella* spp.	0	0	0	0	0	0	0
*P. fluorescens*	0	0	0	0	0	0	0
*P. aeruginosa*	0	0	1	0	0	0	0
*Acinetobacter* spp.	0	1	0	0	0	0	0
*Ralstonia pickettii*	0	0	0	0	0	0	0

Red—bacterial strains from acute/acute hematogenous PJIs; D—days; *S.*—*Staphylococcus*; MRSA—Methicillin-resistant *S. aureus*; MSSA—Methicillin-susceptible *S. aureus; Str.*—*Streptococcus*; *Ent.*—*Enterococcus; E.*—*Escherichia; P.*—*Pseudomonas*.

**Table 3 jcm-11-02238-t003:** Isolated bacterial species (no. of strains identified) from synovial fluid/incubation period days.

Microorganisms	No. of Positive Cultures (No. of Strains Identified) from Synovial Fluid/Incubation Period Days						
	D 1	D 2	D 3	D 4	D 5–D 7	D 8	D 9–D 14
S. epidermidis	0	** 1 **	** 1 **	0	0	0	0
*S. lentus*	0	0	0	1	0	0	0
*S. xylosus*	0	0	0	0	0	0	0
MRSA	** 1 **	1	1	1	0	0	0
MSSA	0	0	1	0	0	0	0
*Str.* group D	0	0	0	0	0	0	0
*Ent. faecalis*	0	0	0	0	0	0	0
*E. coli*	0	0	0	0	0	0	0
*Enterobacter cloacae complex*	0	0	0	0	0	0	0
*Enterobacter amnigenus 2*	0	0	0	0	0	0	0
*Klebsiella* spp.	0	0	0	0	0	0	0
*P. fluorescens*	0	0	0	0	0	0	0
*P.aeruginosa*	0	0	0	0	0	0	0
*Acinetobacter* spp.	0	0	0	0	0	0	0
*Ralstonia pickettii*	0	0	0	0	0	0	0

Red—bacterial strains from acute/acute hematogenous PJIs; D—days; *S.*—*Staphylococcus*; MRSA—Methicillin-resistant *S. aureus*; MSSA—Methicillin-susceptible *S. aureus*; *Str.*—*Streptococcus*; *Ent.*—*Enterococcus*; *E.*—*Escherichia*; *P.*—*Pseudomonas*.

## Data Availability

All data generated or analyzed during this study are included in this published article.

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
