# Peer review of "Positivity Trends of Bacterial Cultures from Cases of Acute and Chronic Periprosthetic Joint Infections"

_jcm, 2022, doi:10.3390/jcm11082238_

Round 1

Reviewer 1 Report

Birlutiu et al., prospectively studied the positivity trends of bacterial cultures from patients who underwent revision surgery for acute and chronic periprosthetic joint infections. Of the included forty cases, 11 were acute/acute hematogenous PJIs, and remaining were chronic PJIs. In acute and chronic cases of PJIs, all bacterial isolates were identified within 3 and 8-days respectively. Authors concluded that for chronic cases of PJI, extend incubation of bacterial cultures are required but not more than 14-days. 

I have few comments or suggestions for the authors to consider:

  1. Although the topic is of interest, English language used in the manuscript needs to be polished.
  2. Line 25-27 (abstract): Since there are few other methods also reported to improve the bacteriological diagnosis of PJI, rephrase the sentence to something like “Sonication fluid cultures increase the positivity rate and helped in identifying rare pathogens like Ralstonia pickettii from chronic PJIs but also ……
  3. Line 37-41 (Introduction): Rephrase the sentence
  4. Use the abbreviations correctly and where ever possible in the manuscript.
  5. Introduction: Authors needs to briefly describe the reported publications on the topic of research and highlight the novelty of the study.
  6. Line 65-66: No clarity in the secondary aim. Please elaborate or delete.
  7. Line 67-70: I don’t think this paragraph is relevant for introduction section.
  8. Materials and methods: Avoid repetition of sentences and make it easy for the readers to follow the article. Authors needs to clarify which classification they have used for differentiating acute/acute hematogenous and chronic PJIs. Details of statistical analysis should be placed at the end of M & M section.
  9. Line 148-166 (results):  It would be easy for the reader if authors provide a flow diagram showing the details of the enrolled patients, including how many of them were acute/chronic PJI.
  10. Line 160-166: Do not confuse the readers. Give the details of the result most essential for the article.
  11. Line 170-171: Avoid repetition.
  12. Table 1-3: Use abbreviation “D” for days so that top row may look better. All microorganisms details in column 1 should be aligned to “Left” and use abbreviations wherever possible (e.g. MRSA/MSSA)
  13. Figure 3: Letters are too small, so replace the image with a high resolution image.
  14. Line 248-251: To denote the statistical significance, use “p” instead of “sig”.
  15. Discussion: Rather than quoting references, authors needs compare and discuss their findings with the previous publications.
  16. Look for other relevant articles that are worth of discussing in the context of present study. E.g. Schwotzer N, Wahl P, Fracheboud D, Gautier E, Chuard C, Patel R. 2014.Optimal culture incubation time in orthopedic device-associated infec-tions: a retrospective analysis of prolonged 14-day incubation. J ClinMicrobiol 52:61–
  17. Study limitations: Please rewrite the limitations part. Is the study retrospective or prospective?
  18. Line 347-349: Delete.

Author Response

Reviewer 1

Birlutiu et al., prospectively studied the positivity trends of bacterial cultures from patients who underwent revision surgery for acute and chronic periprosthetic joint infections. Of the included forty cases, 11 were acute/acute hematogenous PJIs, and remaining were chronic PJIs. In acute and chronic cases of PJIs, all bacterial isolates were identified within 3 and 8-days respectively. Authors concluded that for chronic cases of PJI, extend incubation of bacterial cultures are required but not more than 14-days.

I have few comments or suggestions for the authors to consider:

Answer: Thank you for taking from your precious time to be able to assess our manuscript. The comments were highly insightful and enabled us to improve our manuscript.

Although the topic is of interest, English language used in the manuscript needs to be polished.

A: We hope that we were able to improve the English Language of our manuscript. Thank you.

Line 25-27 (abstract): Since there are few other methods also reported to improve the bacteriological diagnosis of PJI, rephrase the sentence to something like “Sonication fluid cultures increase the positivity rate and helped in identifying rare pathogens like Ralstonia pickettii from chronic PJIs but also ……

A: Thank you for the suggestion. Done!

Line 37-41 (Introduction): Rephrase the sentence

A: Thank you for the suggestion. Done!

Use the abbreviations correctly and where ever possible in the manuscript.

A: Thank you for the suggestion and for pointing this fact. Done!

Introduction: Authors needs to briefly describe the reported publications on the topic of research and highlight the novelty of the study.

A: Thank you for the suggestion and for pointing this fact. We hope that we were able to highlight more the novelty of the study.

Line 65-66: No clarity in the secondary aim. Please elaborate or delete.

A: We rephrased the sentence, hope that it is clear now.

Line 67-70: I don’t think this paragraph is relevant for introduction section.

A: As suggested this paragraph has been deleted from the introduction section of the manuscript.

Materials and methods: Avoid repetition of sentences and make it easy for the readers to follow the article. Authors needs to clarify which classification they have used for differentiating acute/acute hematogenous and chronic PJIs. Details of statistical analysis should be placed at the end of M & M section.

A: We hope that we clarified the issue with classification with the details that we added into the materials and methods section and in the results section of the manuscript. The statistical analysis section of the manuscript was places, as suggested, at the end of the materials and methods section.

Line 148-166 (results):  It would be easy for the reader if authors provide a flow diagram showing the details of the enrolled patients, including how many of them were acute/chronic PJI.

A: Thank you for the suggestion. We included in our manuscript a flow diagram. Done!

Line 160-166: Do not confuse the readers. Give the details of the result most essential for the article.

A: We hope that we are not confusing the readers and we report only the results that are essential and should be reported.

Line 170-171: Avoid repetition.

A: As suggested this section has been deleted from the manuscript.

Table 1-3: Use abbreviation “D” for days so that top row may look better. All microorganisms details in column 1 should be aligned to “Left” and use abbreviations wherever possible (e.g. MRSA/MSSA)

A: We performed the changes as suggested.

Figure 3: Letters are too small, so replace the image with a high resolution image.

A: Done!

Line 248-251: To denote the statistical significance, use “p” instead of “sig”.

A: Done!

Discussion: Rather than quoting references, authors needs compare and discuss their findings with the previous publications. Look for other relevant articles that are worth of discussing in the context of present study. E.g. Schwotzer N, Wahl P, Fracheboud D, Gautier E, Chuard C, Patel R. 2014.Optimal culture incubation time in orthopedic device-associated infec-tions: a retrospective analysis of prolonged 14-day incubation. J ClinMicrobiol 52:61–

A: Thank you for the suggestion.

Study limitations: Please rewrite the limitations part. Is the study retrospective or prospective?

A: We reassessed the study limitations section of the manuscript. It is a retrospective analysis of prospectively collected data.

Line 347-349: Delete.

A: We performed the changes as suggested.

We hope that the revised form of the manuscript and our accompanying responses will be sufficient to make our manuscript suitable.

Reviewer 2 Report

Dear Authors,

thank you ver much fort he opportunity to review your manuscript submitted to Journal of clinical medicine.

The authors present a prospective data set on micorbiological findings in more then 40 patients with revised joint arthroplasties from a single center in Rumania collected between 2016 and 2019.

The focus their studyy on the description of bacteriological findings in cultures and comparison of different sample types and time to detection of growth. They additionally subanalyse these factors for the different types of infections.

The basic idea is good and novel to a certain degree. However, the execution is not satisfactory, and neither is the interpretation of the findings. In my opinion there a some key aspects which need to be reworked extensively.

In General it is very difficult to get information how patients were included and excluded frome the analysis. Could you name the criteria and describe the process? Could you you also name the number of patients that were excluded and for what reasons.

The cohort is missing information on the criteria of the individual variables oft he MSIS criteria used to proof PJI. In general it is not coheren if you used the Zimmerli definition of PJI, the Pro Implant definition.

Can you add definitions how you interpreted culture results? What did you do when only 1 positives culture occured ? Your definition: „1 positive culture combined with 133 an additional other positive criteria, for example, an increased leukocyte count in synovialfluid or positive intraoperative histopathological examination“ is tricky and could cause flase positives to be included into your dataset. How did account for potential contamination in sonication (R. pickettii). Did you use sonication with  cutoff values (e.q. 50 CFU/ml). How did you perform culturing (which media? Enrichment beside Blood culture? Lenght of incubation for each).

I think it would be more interessting tot he reader to see how the different tests and culturing approaches perform against each other , especially if you have a high adherence to your sampling strategy.

Your discussion lacks potential inclusion of false positive culture results ( known contaminants of Sonication such as R. pickettii).

Your Limitations section lacks a critical evaluation of your data set and limitations of the methodology used.

There are a considerable amount of typos as well as issues with phrasing and grammer. You should consider editing services from a native speaker.

Author Response

Reviewer 2

Dear Authors,

thank you ver much fort he opportunity to review your manuscript submitted to Journal of clinical medicine. The authors present a prospective data set on micorbiological findings in more then 40 patients with revised joint arthroplasties from a single center in Rumania collected between 2016 and 2019. The focus their studyy on the description of bacteriological findings in cultures and comparison of different sample types and time to detection of growth. They additionally subanalyse these factors for the different types of infections. The basic idea is good and novel to a certain degree. However, the execution is not satisfactory, and neither is the interpretation of the findings. In my opinion there a some key aspects which need to be reworked extensively.

Answer: Thank you for taking from your precious time to be able to assess our manuscript. The comments were highly insightful and enabled us to improve our manuscript.

In General it is very difficult to get information how patients were included and excluded frome the analysis. Could you name the criteria and describe the process? Could you you also name the number of patients that were excluded and for what reasons.

A: Thank you for the suggestion and for pointing this fact. For this reason, we included in our manuscript a flow diagram so that it would be easy for the reader to review the details of the enrolled patients, including how many of them were acute/chronic PJI.

The cohort is missing information on the criteria of the individual variables oft he MSIS criteria used to proof PJI. In general it is not coheren if you used the Zimmerli definition of PJI, the Pro Implant definition.

A: We apologies for not being clear with our methods. Periprosthetic joint infection was defined using the criteria from the workgroup of the Musculoskeletal Infection Society published by Parvizi et al. We used the classification proposed by Zimmerli et al. and also the classification from the Pocket Guide to Diagnosis & Treatment of Periprosthetic Joint Infection (PJI) of the PRO-IMPLANT Foundation, Berlin, Germania (coordinated by N. Renz and A. Trampuz) a guide that is in line with national and international recommendations and that defines periprosthetic infections as acute or chronic (Perioperative/Hematogenous or per continuitatem). We reported our data to both classifications. We also added into the results section of the manuscript that all the results are reported to the classification proposed by the PRO-IMPLANT Foundation. Thank you!

Can you add definitions how you interpreted culture results? What did you do when only 1 positives culture occured ? Your definition: „1 positive culture combined with 133 an additional other positive criteria, for example, an increased leukocyte count in synovialfluid or positive intraoperative histopathological examination“ is tricky and could cause flase positives to be included into your dataset. How did account for potential contamination in sonication (R. pickettii). Did you use sonication with  cutoff values (e.q. 50 CFU/ml). How did you perform culturing (which media? Enrichment beside Blood culture? Lenght of incubation for each).

A: Thank you for the pointing this issue and for giving us the possibility to make some discussion on this matter. In our opinion there is no contamination, as R. pickettii was isolated from the sonication fluid. Being a Gram-negative bacilli, non-fermentive, oxidase-positive bacteria, it can be confused with other bacteria like Pseudomonas aeruginosa, the isolated strain and a sample of the sonication fluid was shipped to a reference laboratory were molecular diagnostic teste were used and conformed our results (R. pickettii was isolated). Never the less, a contamination of the container in which the implant is shipped for sonication and then to the laboratory is not possible as the implant is inserted in the container under strict sterile conditions and the Ringer’s or saline sterile solution is added in the same manner, the next time when the container is open is at the laboratory wherec onication fluid is harvest for the cultures and for the rapid assay, the container is opened in a biological safety cabinet, and also the operating rooms where joint arthroplasties are performed are equipped with high efficiency air filtration systems, by and also based on the fact that regular epidemiological surveillance is conducted in the operating theater with samples that are collected form the surfaces, appliances, air filters, solutions, and so on. In this setting the PJIs could possibly be associated with the saline lavage used during the initial total joint arthroplasty – on this matter on a regular base samples from this solution are harvest as a mean of prevention, unfortunately form the period of the TJA there is no data on this matter. There are not so many reports of PJIs caused by R. pickettii published in the literature but based on the above statements, our experience also previously published, our results there is no doubt that R. pickettii the causative agent of the PJIs.

Based on the used criteria and with our last analysis published last year, our protocol gives us good results. As it is not part for this manuscript we also need to mention the use of a bbFISH kit in the assessment of the sonication fluid.

The sonication fluid cultures were considered positive, if >50 CFU/ml of sonication fluid were counted. We also added the statement in the manuscript.

As from the discussions from our CM for the sonication fluid the resulted precipitate is inoculated onto Columbia agar with sheep blood (incubated aerobically, anaerobically and in high concentration of CO2), Sabouraud plate, MacConkey agar plate, chocolate agar, glucose broth, lactose broth, thioglycollate broth, brucella blood agar and fastidious broth.

I think it would be more interessting tot he reader to see how the different tests and culturing approaches perform against each other , especially if you have a high adherence to your sampling strategy.

A: Unfortunately, this was not the aim of this study, thank you for the suggestion for a future manuscript, we published a couple of year ago some data on this matter.

Unpublished data:

For periprosthetic tissue cultures, sensibility 80.65%, specificity100.00%

For bbFISH, sensibility 83.87%, specificity100.00%

For sonication fluid cultures, sensibility 90.32%, specificity100.00%

For synovial fluid cultures, sensibility 32.26%, specificity100.00%

Your discussion lacks potential inclusion of false positive culture results (known contaminants of Sonication such as R. pickettii).

A: Thank you for the pointing this issue and for giving us the possibility to make some discussion on this matter. In our opinion there is no contamination, as R. pickettii was isolated from the sonication fluid. Being a Gram-negative bacilli, non-fermentive, oxidase-positive bacteria, it can be confused with other bacteria like Pseudomonas aeruginosa, the isolated strain and a sample of the sonication fluid was shipped to a reference laboratory were molecular diagnostic teste were used and conformed our results (R. pickettii was isolated). Never the less, a contamination of the container in which the implant is shipped for sonication and then to the laboratory is not possible as the implant is inserted in the container under strict sterile conditions and the Ringer’s or saline sterile solution is added in the same manner, the next time when the container is open is at the laboratory where sonication fluid is harvest for the cultures and for the rapid assay, the container is opened in a biological safety cabinet, and also the operating rooms where joint arthroplasties are performed are equipped with high efficiency air filtration systems, by and also based on the fact that regular epidemiological surveillance is conducted in the operating theater with samples that are collected form the surfaces, appliances, air filters, solutions, and so on. In this setting the PJIs could possibly be associated with the saline lavage used during the initial total joint arthroplasty – on this matter on a regular base samples from this solution are harvest as a mean of prevention, unfortunately form the period of the TJA there is no data on this matter. There are not so many reports of PJIs caused by R. pickettii published in the literature but based on the above statements, our experience also previously published, our results there is no doubt that R. pickettii the causative agent of the PJIs.

We added a section in the manuscript.

Your Limitations section lacks a critical evaluation of your data set and limitations of the methodology used.

A: The limitation section of the manuscript was reassessed. Thank you!

There are a considerable amount of typos as well as issues with phrasing and grammer. You should consider editing services from a native speaker.

A: We hope we addressed the mentioned issues. Thank you.

We hope that the revised form of the manuscript and our accompanying responses will be sufficient.

Reviewer 3 Report

Thanks for the opportunity to review the manuscript entitled “positivity trends of bacterial cultures from cases of acute and chronic periprosthetic joint infections”. The authors retrospectively collected and analyzed the patients’ data with acute or chronic PJI, especially the incubation time to detect some microorganisms. The results found it would take mean 3 days to identify specific strains of acute/ acute hematogenous PJI, and 8 days for chronic PJIs. In my opinion, this study is designed and written well. Some minor issues should be addressed.

  1. p1, line 23, the sum of 11 acute and 38 chronic PJI is not 40.

Author Response

Reviewer Comments:

Reviewer 3

Thanks for the opportunity to review the manuscript entitled “positivity trends of bacterial cultures from cases of acute and chronic periprosthetic joint infections”. The authors retrospectively collected and analyzed the patients’ data with acute or chronic PJI, especially the incubation time to detect some microorganisms. The results found it would take mean 3 days to identify specific strains of acute/ acute hematogenous PJI, and 8 days for chronic PJIs. In my opinion, this study is designed and written well. Some minor issues should be addressed.

Answer: Thank you for taking from your precious time to be able to assess our manuscript. The comments were highly insightful and enabled us to improve our manuscript.

p1, line 23, the sum of 11 acute and 38 chronic PJI is not 40.

A: Thank you for pointing this fact. We addressed the issue. We apologies for the typing mistake.

We hope that the revised form of the manuscript and our accompanying responses will be sufficient.

Round 2

Reviewer 1 Report

  1. Page 4, line 181: "All our results will be reported at this classification". Correct the grammatical mistake.
  2. Authors needs to carefully check the entire manuscript for grammatical mistakes.
  3. Line 348-367: Since this study is not a case report, authors should not describe extensively about the isolation of R. pickettii. So delete lines 348-367.
  4. Line 381-382: Delete "is difficult". 

Good Luck!

Author Response

Sibiu, 27.03.2022

To

the Editors of Journal of Clinical Medicine®

Dear Editor,

Dear reviewer,

Thank you for reviewing our manuscript. Please find attached a revised version of our manuscript, “Positivity trends of bacterial cultures from cases of acute and chronic periprosthetic joint infections.”.

Yours and the reviewers’ comments were highly insightful and enabled us to greatly improve the quality of our manuscript. We have modified the manuscript in response to the comments. Attached are our point-by-point response to each comment.

Reviewer Comments:

Reviewer 1

Page 4, line 181: "All our results will be reported at this classification". Correct the grammatical mistake.

Authors needs to carefully check the entire manuscript for grammatical mistakes.

Line 348-367: Since this study is not a case report, authors should not describe extensively about the isolation of R. pickettii. So, delete lines 348-367.

Line 381-382: Delete "is difficult".

Good Luck!

Answer: Thank you for taking from your precious time again to be able to assess our revised manuscript. The comments were highly insightful. We hope that we were able to improve the English Language of our manuscript, by polishing the manuscript using a dedicated editing service. The paragraph regarding R. pickettii was added to the manuscript as one of the other reviewers questioned the isolation of this strain and requested discussions on the potential inclusion of false positive culture results (known contaminants of sonication such as R. pickettii). The suggested words were deletes. We also addressed the grammatical mistake.

We hope that the revised form of the manuscript and our accompanying responses will be sufficient to make our manuscript suitable and accepted for publication in Journal of Clinical Medicine®. We shall look forward to hearing from you at your earliest convenience.

With our best regards,

Sincerely yours,

Rares Mircea Birlutiu, MD PhD

Victoria Birlutiu, Assoc. Prof. MD. PhD

Reviewer 2 Report

I feel that my remarks were not sufficiently addressed.

In general the main problem with the revised version remains the flawed definition of PJI. If you use the 2018 MSIS ICM definition, then the 1 culture positive + one more criteria (which you do not explicitely name) will not add up to a proven PJI but only to a possible PJI. In addition the centrifugation with the addition of a 50CFU/ml is not a validated procedere. You either use centrifugation or the cutoff. This lead to a reduced specificity of the definitions. 

The revised parts show several typos. Grammer issues remain present throughout the entire manuscript. Please consider the help of a native speaker.

Author Response

Sibiu, 27.03.2022

To

the Editors of Journal of Clinical Medicine®

Dear Editor,

Dear reviewer,

Thank you for reviewing our manuscript. Please find attached a revised version of our manuscript, “Positivity trends of bacterial cultures from cases of acute and chronic periprosthetic joint infections.”.

Yours and the reviewers’ comments were highly insightful and enabled us to greatly improve the quality of our manuscript. We have modified the manuscript in response to the comments. Attached are our point-by-point response to each comment.

Reviewer Comments:

Reviewer 2

I feel that my remarks were not sufficiently addressed.

In general the main problem with the revised version remains the flawed definition of PJI. If you use the 2018 MSIS ICM definition, then the 1 culture positive + one more criteria (which you do not explicitely name) will not add up to a proven PJI but only to a possible PJI. In addition the centrifugation with the addition of a 50CFU/ml is not a validated procedere. You either use centrifugation or the cutoff. This lead to a reduced specificity of the definitions.

The revised parts show several typos. Grammer issues remain present throughout the entire manuscript. Please consider the help of a native speaker.

Answer: Thank you for taking from your precious time again to be able to assess our revised manuscript. The comments were highly insightful. We apologies that you fell that your remarks were not sufficiently addressed, we took great time to be able to address them to our best knowledge. We hope that we were able to improve the English Language of our manuscript, by polishing the manuscript using a dedicated editing service. We revised the PJI criteria section of the manuscript, we hope it is clearer. The decision to harvest 50 mL of sonication fluid after sonication and to centrifuge them at 2500 rpm for 5 minutes and then to inoculate the resulting precipitate was made by our team of CM form the lab, and also the >50 CFU/ml cutoff value. We know that, using a cutoff of ≥50 CFU/ml, sonication shows higher sensitivity than vortexing that if why we do not use vortexing in our lab, while the specificities remain more or less equal. Using a lower cutoff value (of ≥1 CFU/ml), will provide similar sensitivities of vortexing and sonication but with a decreased specificity. As data from the literature show. Again, we do not use vortexing.

We hope that the revised form of the manuscript and our accompanying responses will be sufficient to make our manuscript suitable and accepted for publication in Journal of Clinical Medicine®. We shall look forward to hearing from you at your earliest convenience.

With our best regards,

Sincerely yours,

Rares Mircea Birlutiu, MD PhD

Victoria Birlutiu, Assoc. Prof. MD. PhD